# The Influence of Selected Gastrointestinal Parasites on Apoptosis in Intestinal Epithelial Cells

**DOI:** 10.3390/biom10050674

**Published:** 2020-04-27

**Authors:** Patrycja Kapczuk, Danuta Kosik-Bogacka, Patrycja Kupnicka, Emilia Metryka, Donata Simińska, Karolina Rogulska, Marta Skórka-Majewicz, Izabela Gutowska, Dariusz Chlubek, Irena Baranowska-Bosiacka

**Affiliations:** 1Department of Biochemistry and Medical Chemistry, Pomeranian Medical University, Powstańców Wlkp. 72, 70-111 Szczecin, Poland; patrycja2510@o2.pl (P.K.); patrycjakupnicka@o2.pl (P.K.); emilia_metryka@o2.pl (E.M.); d.siminska391@gmail.com (D.S.); dchlubek@pum.edu.pl (D.C.); 2Department of Biology and Medical Parasitology, Pomeranian Medical University, Powstańców Wlkp. 72, 70-111 Szczecin, Poland; kodan@pum.edu.pl; 3Department of Microbiology, Immunology and Laboratory Medicine, Pomeranian Medical University, Powstańców Wlkp. 72, 70-111 Szczecin, Poland; karolina.rogulska20@gmail.com; 4Department of Medical Chemistry, Pomeranian Medical University, Powstańców Wlkp. 72, 70-111 Szczecin, Poland; marta_skorka@o2.pl (M.S.-M.); izagut@poczta.onet.pl (I.G.)

**Keywords:** apoptosis, molecular mechanisms of apoptosis, parasites, gastrointestinal tract

## Abstract

Studies on the parasite–host interaction may provide valuable information concerning the modulation of molecular mechanisms as well as of the host immune system during infection. To date, it has been demonstrated that intestinal parasites may affect, among others, the processes of digestion in the gastrointestinal system of the host, thus limiting the elimination of the parasite, the immune response as well as inflammation. However, the most recent studies suggest that intestinal parasites may also affect modulation of the apoptosis pathway of the host. The present paper presents the latest scientific information on the influence of intestinal parasite species (*Blastocystis* sp., *Giardia* sp., *Cryptosporidium* sp., *Trichuris* sp., *Entamoeba histolytica*, *Nippostrongylus brasiliensis*, *Heligmosomoides polygyrus*) on the molecular mechanisms of apoptosis in intestinal epithelial cells. This paper stresses that the interdependency between the intestinal parasite and the host results from the direct effect of the parasite and the host’s defense reactions, which lead to modulation of the apoptosis pathways (intrinsic and extrinsic). Moreover, the present paper presents the role of proteins involved in the mechanisms of apoptosis as well as the physiological role of apoptosis in the host’s intestinal epithelial cells.

## 1. Introduction

Various cases of the modulation of apoptosis in host cells due to the presence of parasites can be found in the literature [1,2,3]. Parasites can inhibit the process of apoptosis in host cells by accelerating the death of immunologically competent cells. Such activity is manifested by *Leishmania* sp. ininhibiting apoptosis in macrophages [4]. Literature on the subject also reports cases of the induction of apoptosis, for example, by *Trypansoma cruzi*, causing activation of the programmed CD4+ and CD8+ lymphocyte death pathways, and due to lymphocyte induction, an increase in the expression of Fas (CD95) and Fas-L antigen protein. Additionally, *T. cruzi* may directly cause T lymphocyte apoptosis through glycosides containing ceramides [1].

The phenomenon of the modulation of apoptotic mechanisms in host enterocytes by intestinal parasites is still under research. To date, only a few studies have been conducted on the stimulation of apoptosis by intestinal parasites.

## 2. Apoptosis

Genetically programmed cell death (i.e., apoptosis) is not just a basic physiological process involved in the proper development and maintenance of homeostasis of an organism, but also takes part in numerous pathological processes in the body such as the elimination of neoplastic cells and those infected with a virus. Theses physiological processes lead to the formation of apoptotic bodies that subsequently undergo phagocytosis. This active and structured process occurs without inflammation or damage to the surrounding tissues, and requires energy input in the form of ATP—in contrast to necrosis. Apoptotic cells are characterized by a loss of water, shrinkage, changes in shape, atrophy, condensation of the cytoplasm and tightly packed organelles, marginalization, and condensation of chromatin as well as fragmentation of genomic DNA by endonucleases [5,6,7]. Apoptosis can be induced by a number of factors such as reactive oxygen species, UV, and ionizing radiation, pro-inflammatory cytokines (tumor necrosis factor TNF, interferon IFN), slight temperature changes, medication, the course of degenerative diseases (AIDS, Parkinson’s disease, Alzheimer’s disease), and infectious diseases such as viral hepatitis [5,8,9,10,11,12].

### 2.1. Proteins Involved in the Mechanisms of Apoptosis

It has been proven that proteins are indispensable for the transmission cascade of apoptotic signals. A key role in apoptosis is played by caspase group enzymes, which occur in the form of inactive zymogens—procaspases (inactive precursors of enzymes). Having received the apoptotic signal, cysteine protease undergoes proteolysis to an active form. These enzymes are elements of the cascade reaction, however, they also play other roles (i.e., regulatory, effector and initiating roles) in the process of apoptosis. Apart from apoptosis, they affect modulation of the inflammatory response as well as proliferation and cell differentiation processes. With respect to their structure (length of the prodomain), three groups of caspases are distinguished: caspases with the so-called death domain (pro-inflammatory caspases), which include caspase-1, -4, -5, -11, -12, -13 and -14; effector (executioner) caspase-3, -6 and -7 with a short domain; and apoptosis initiator caspase-2, -8, -9, and -10 [13,14]. A significant role in the process of apoptosis is played by caspase-3, the activation of which is necessary in all pathways of apoptotic signaling. Other key proteins are inhibitors and activators of caspase in a cell. Since caspases determine cell survival, their activity must be strictly regulated. The factors inhibiting apoptosis include Inhibitor of Apoptosis Protein (IAP): cIAP1, cIAP2, XIAP (X-linked mammalian inhibitor of apoptosis protein), NAIP (neuron al apoptosis inhibitory protein), and heat shock proteins (HSP): Hsp70 and Hsp90 [15,16,17,18].

Another protein that is of vital importance to apoptosis is poly(ADP-ribose) polymerase (PARP), which plays the role of DNA reparatory enzyme in an organism. Activation of this protein occurs when double strands of DNA break. When apoptotic mechanisms are initiated such ascaspase-3 and -7, PARP undergoes inactivation, which consequently results in the inhibition of the protein’s reparatory ability and induction of cell death. Therefore, the inactivated form of PARP (cPARP) is classified as a marker of apoptosis [19].

The process of apoptosis also includes proteins from the Bcl-2 family, showing homology in BH regions (Bcl-2 homology). One of the subfamilies of Bcl-2 includes anti-apoptotic proteins that act as death repressors, and are distinguished by having four BH domains such as Bcl-xL, Bfl-1, Mcl-1, A1, BHRF, and Bcl-2. The function of these proteins is to prevent the release of cytochrome c from the mitochondrion and other apoptogenic factors to the cytoplasm, thus blocking apoptotic signals. The other subfamily of proteins involved in apoptotic pathways are proapoptotic proteins such as Bax, Bak, and Bok, containing three BH domains as well as Bid, Bam, Bim, Bmf containing one BH3 domain. Proapoptotic proteins have various functions that facilitate apoptosis such as infiltrating the mitochondrion membrane and allowing the outflow of apoptotic factors, protecting Bax from being inactivated by antiapoptotic protein, and facilitating oligomerization of the remaining proteins from this subfamily [20,21,22]. Similarly to caspases, proteins from the Bcl-2 family are regulated through modulation of their transcription by p53 protein [23].

Auxiliary functions in apoptotic pathways are performed by NOXA proteins (proapoptotic protein from BH3-only subfamily), PUMA (p53 up regulated modulator of apoptosis), and p53 protein. NOXA proteins are responsible for inhibiting the activity of antiapoptotic proteins, while PUMA proteins are involved in increasing the expression and in changes in Bax protein conformation [24] (Figure 1).

Numerous studies have proven that p53 protein (tumor suppressor protein) is responsible for the induction of apoptosis. By the action of TAD (N-terminal transactivation domain) and TAD2 (second transactivation domain of p53), the p53 protein induces apoptosis without gene transactivation (i.e., by interaction with other proteins (Bcl-XL)). It has also been shown that the p53 protein may attach to apoptotic proteins from the Bcl family (e.g., Bax) and promote oligomerization, which effectively promotes apoptosis. Additionally, it was found that a p53 protein without an active AD2 domain (TAD subdomain with amino acid 1–40 residues) cannot induce apoptosis. Amphipathic motifs within both AD1 and AD2 form a stable helical structure upon binding to target proteins [25].

### 2.2. Pathways of Apoptosis and the Execution Phase

Even though the process of apoptosis always follows a specific pattern, the process itself is very complicated. In individual stages of the process, numerous proteins mediate the cascade reaction from the receptors present on the cell surface to proteolytic enzymes destroying the cytoskeleton. Depending on the type of cell and the external factor inducing the process, there are two basic pathways of apoptosis activation: extrinsic (receptor) and intrinsic (related to mitochondrion).

The extrinsic pathway is determined by the interaction of ligands with membrane receptors belonging to the superfamily of tumor necrosis factor (TNF) receptors: TNFR1, TNFR2, Fas/CD95/Apo1 and ligands TRAIL (TNF-related apoptosis-inducing ligand)/Apo2. The TNF receptor is composed of an expanded 80-amino acid death domain (TRADD-associated death domain), and domains rich in cysteine. A cell receives an external signal that initiates the mechanism of apoptosis. As a result of attaching the ligand to the receptor, TNFR1 is activated. Due to conformational changes, the death domain is released to the cytoplasm, which takes part in the formation of complexes I and II in the apoptosis pathways. Similar to TRADD, complex I comprises receptor-interacting protein 1 (RIP1) and TNF-receptor associated factor 2 (TRAF2). As a result of the formation of complex I, the following are activated: receptor-interacting serine/threonine-protein kinase 1 (RIPK1), nuclear factor kappa B (NF-KB), and protein kinases: c-Jun N-terminal *kinases* (JNK) and p38 protein. Similar to complex I, complex II is composed of TRADD, Fas-associated death domain (FADD), and caspase-8. Subsequently, the signal is transmitted to adaptor protein FADD, which connects to the death effector domain (DED) to the DED section of the initiating procaspase. Along with activation of the extrinsic-origin pathway of apoptosis with the formation of complexes, there is also the activation of TNFR1 through binding the TRADD domain by RIP (Receptor-interacting protein) and direct activation of caspase-2. The cascade reaction of the executioner caspases is triggered both through these complexes as well as through the alternative pathway. The final stage of active caspase 8 is the activation of proapoptotic protein Bid to tBid (active form). As a consequence of tBid formation, there is an interaction with proapoptotic proteins Bax/Bak, which contributes to increased permeability of the mitochondrial outer membrane. Consequently, the initiation of cell death is promoted [5,8,9,26,27,28,29].

The intrinsic pathway of apoptosis is generally caused by damage to the organelles: mitochondria, lysosomes, endoplasmic reticulum, and Golgi apparatus. This pathway is also activated by an increased concentration of reactive oxygen species (ROS), DNA damage, oxidative stress, increased concentration of Ca^2+^ ions, and disturbed electrolyte transport as well as cytotoxic medicines and pathogens. Regulation of the intrinsic pathway is connected with the disturbed balance in active proapoptotic proteins and antiapoptotic proteins inactivated by phosphorylation. Increased accumulation of proapoptotic proteins results in enlargement of the pores in the mitochondrion membrane and destabilization due to oligomerization of mostly the Bax and Bak proteins. In natural conditions, proapoptotic proteins (Bax, Bak) are inactive due to binding with antiapoptotic proteins Bcl-2 and Bcl-XL. Consequently, these factors cause increased permeability of the mitochondrion membrane, which in turn leads to disturbances in the membrane potential and a release of apoptogenic factors: cytochrome c (Apaf-2), Smac (Second mitochondria-derived activator of caspase)/DIABLO (direct inhibitor of apoptosis-binding protein with low pI), HtrA2/Omi (serine protease, an antagonist of IAPs—high temperature requirement A2) and AIF (Apoptosis Inducing Factor) to the cytoplasm. Additionally, the production of ATP and glutathione is decreased. Due to biochemical reactions, an excessive number of reactive oxygen species from the conversion of NADH and NADPH into the oxidized form is observed. Electron-transporting proteins such as cytochrome c take part in oxidative phosphorylation and the production of ATP. Moreover, cytochrome c, which attaches to the outer surface of the internal membrane of the mitochondrion, is released along with negatively charged cardiolipin to the cytosol. Subsequently, it connects with adaptor protein Apaf-1 (apoptotic protease activating factor -1), then with caspase-9 and dATP forming apoptosome. As a result of activation of the effector caspases, apoptosis takes place. Smac/DIABLO and HtrA2/Omi proteins are characterized by an inhibitory effect on the apoptosis inhibitory proteins. Additionally, they may have an indirect effect on caspase-3 activation and thus promote the apoptosis pathway. Due to an affinity with the cell nucleus, AIF protein causes DNA fragmentation [30,31].

Apart from the aforementioned pathways, there are other signaling pathways of apoptosis: the pseudo-receptor pathway occurring in T lymphocytes, NK cells (Natural Killer), and neutrophils with the involvement of perforins and granzyme B; and the ceramide/sphingomyelin pathway or stress-induced pathway [8].

The final stage of the whole process of apoptosis is the execution stage. It is marked by the high activity of effector caspase, leading to degradation of the membrane of the cytoskeleton and nucleus as well as cytokeratin and PARP. A crucial role is played by caspase-3, which is responsible for releasing CAD endonuclease from the complex with an inhibitor (ICAD). Then, the activated CAD leads to degradation of DNA and condensation of the chromatin. Additionally, caspase-3 leads to the development of apoptotic bodies, on the surface of which is phosphatidylserine, which in turn is indispensable for the recognition by phagocytes. The rapid uptake of apoptotic bodies by phagocytes prevents inflammatory reactions by the limited release of cell disintegration components into the extracellular space [32].

### 2.3. Physiological Role of Apoptosis in the Intestinal Epithelium

In its physiological state, apoptosis is a process aimed at maintaining the balance in tissues and organs, and is characterized by a high rate of cell division such as in the cells of the external layer of the intestinal epithelium. These cells are characterized by the ability to be shed and undergo atrophy. Then, human intestinal epithelial cells (IECs) are replaced by the cells from the deeper layers of the epithelium, which are pushed toward the surface. In the case of the small intestine epithelium, the process can last from three to five days, and cell proliferation takes place in the intestinal crypts whereas the shedding occurs in the intestinal lumen. The phenomenon of “anoikis” (death by apoptosis) is a consequence of the lack of adhesion to neighboring cells, and a disturbed expression of proteins such as integrins and cadherins. Similar to the mechanism of apoptosis itself, cytokines determine the proliferation and differentiation of intestinal epithelial cells. Cytokines of the transforming growth factor TGFα family take part in cell proliferation and differentiation, while TNF-α and INF-γ cytokines determine apoptosis. In such cases, TNF-α cytokine may constitute both an inductor and inhibitor of apoptosis (Figure 2). When TNF-α binds with the TNFR1 receptor, it can act in higher parts of the villi causing apoptosis, while its activity through the TNFR2 receptor in the intestinal crypts may prevent excessive apoptosis and affect a high level of protein p53. This mechanism can be evidenced by high levels of antiapoptotic protein Bcl-2 in the intestinal crypts, and of Bax in the higher level of villi. It has been demonstrated that the physiological level of apoptosis in the large intestine is lower than that in the small intestine. It is believed that it partly affects a high level of antiapoptotic BCL-2 particles at the crypt base [33,34,35,36].

The physiological role of the mechanisms of apoptosis occurring in the intestine is one of the functions of the epithelial barrier and, consequently, a significant aspect of the protection of mucosa. The intestinal epithelial cells constitute the first line of defense of the host’s cells against external factors. Modulation of apoptotic processes in the intestines, characterized by excessive programmed cell death or disturbed cell division, is associated with numerous diseases of the gastrointestinal tract such as ulcerative colitis or Crohn’s disease. Disturbance of the physiological processes of apoptosis can contribute to or aggravate damage to the mucosa. In the case of bacterial, viral, and parasitic diseases, the primary defense is provided by enterocytes that can stimulate the immune system through the production of mediators, which have the potential to activate the apoptosis pathway in the intestines [35,37].

## 3. The Effect of Intestinal Parasites on Apoptosis in Intestinal Epithelial Cells

There are numerous studies on the characteristics of the mechanisms regulating the apoptosis pathway by viral and bacterial pathogens [37]. Additionally, an ever increasing number of published literature items indicates the existence of an interaction between physiological apoptosis of intestinal epithelial cells and the presence of intestinal parasitic diseases. It has been proven that intestinal parasites can have a multidimensional influence on the molecular mechanisms involved in the parasite–host system [38,39]. These studies have mainly been aimed at understanding the dependence between apoptosis of the host cells and medically relevant parasites, mostly those resulting in human parasitosis. Due to the fact that there are “new” parasite species found in areas so far uninhabited by them as well as the presence of several opportunistic species, infections with theses parasites have been reported. As a result of human activity, some parasite species have acquired the ability to cross natural barriers (e.g., geographical) and to acclimatize in new environments. The presence of non-indigenous species can disturb the parasite–host balance, which consequently may entail a threat to a host [40,41,42]. The efficient immune system of a human is able to protect the organism against the effects of a parasitic infection. However, every disturbance of the immune system has an influence on the course of the infection. In such a case, the course of parasitic infection is often chronic, dynamic, and causes complications [43].

The main characteristic of the intestinal epithelial barrier is the ability to repair the damaged mucosa. The integrity of the mucous membrane of the intestine is determined by its constant proliferation, migration, and enterocyte differentiation. Therefore, each case of a disturbed functioning of the cells such as parasitic infections may contribute to disturbing intestinal homeostasis. Hence, apoptosis plays a vital role in the correct functioning of the intestinal barrier of a host. On one hand, apoptosis performs a protective function: the impermeability of the intestinal epithelial barrier. On the other hand, the parasites use host apoptosis mechanisms to increase intestinal epithelial permeability in order to cause parasitosis. The results published over the last few years demonstrate the effect of species such as *Blastocystis* sp., *Giardia* sp., *Cryptosporidium* sp., *Trichuris* sp., *Entamoeba histolytica*, *Nippostrongylus brasiliensis*, and *Heligmosomoides polygyrus* on apoptosis in the intestines of the host (Figure 3).

### 3.1. Blastocystis sp.

*B. hominis* is a cosmopolitan protozoan affecting both immunologically competent and immunocompromised humans. Other possible hosts could be rats, pigs, and chickens. This parasite is classified as an obligate anaerobe, present in the large intestine of humans. It is characteristic for this parasite to have several morphological forms: vacuolar (Figure 4), non-vacuolar, multi-vacuolar, granular, ameboid, and cyst. In parasitological diagnostics, the differentiation between the morphological forms of *B. hominis* is of critical importance for the correct identification of the parasite. Infection may occur via the fecal–oral route as a result of ingesting a cyst with contaminated food or water. The pathogenicity of *B. hominis* increases in undernourished and oncological patients as well asin those after organ transplants. This protozoan can cause diarrhea, abdominal pain, flatulence, and lack of appetite as well as irritable bowel syndrome, hepatomegaly, splenomegaly, rash, and weight loss. Usually, *B. hominis* infection is asymptomatic [44,45,46,47].

Even though the parasite was discovered a very long time ago and has been studied for many years, there is little data on the pathogenic effects of various *Blastocystis* species on the cells of the host. It has been proven that in some cases, *Blastocystis* sp. can cause an invasion of the intestinal epithelium and damage to the barrier both in vitro and in vivo [48,49,50,51,52,53,54]. Therefore, it is interesting in what way *Blastocystis* can influence apoptosis in host cells. Recently, it has been noted that this parasite can induce the process of caspase-dependent apoptosis without changes in the intestinal epithelium of rats, while at the same time maintaining its correct functioning and impermeability of the barrier [55]. The study by Puthia et al. 2006 [53] showed that the isolate WR1 *B. ratti* induces apoptosis in cells IEC-6 (non-transformed cell line of rat intestinal epithelium), changes the distribution of F-actin, decreases the transepithelial electrical resistance (TER), and increases the permeability of the intestinal epithelium. Various methods and techniques were used in the experiment to identify the mechanisms of apoptosis and its activation including DAPI staining, the annexin V-FITC (fluorescein isothiocyanate) binding test, TUNEL (Terminal deoxynucleotidyl transferase (TdT) dUTP Nick-End Labeling), and the caspase-3 fluorescent test. Changes characteristic for the first stage of apoptosis were identified—chromatin condensation and fragmentation of the cell nucleus—and compared with the control. The early stage of apoptosis was also confirmed with a phosphatidylserine externalization test. To confirm the assumptions of the effect of *Blastocystis* on host cell death without direct contact, Millicell-HA filters were used to separate live parasites from host cells. In this way, it was possible to demonstrate that apoptosis can be induced not only by the mere presence of the parasite, but also by its secretory and non-secretory cellular components. The late stage of cell death (DNA fragmentation) was confirmed by means of the TUNEL technique. In comparison with the control, a significantly increased activity of caspase-3 in IEC-6 cells was found. The study also revealed that treatment with metronidazole could be an antiprotozoal drug for *B. ratti*. Subsequently, this has a positive effect of the functioning of the epithelial membrane as the drug can induce *Blastocystis* cell death. The results by Puthia (2006) suggest that metronidazole has therapeutic potential in *Blastocystis* infection [53]. Additionally, Wu (2014) [54] demonstrated that this parasite is able to induce apoptosis in the human colon cell line Caco-2 (ATCC) using two *Blastocystis* subtypes: ST-7 (isolate B) obtained from a patient with symptoms of blastocytosis in Singapore General Hospital, and ST-4 (isolate WR-1) isolated from a Winstar rat [55]. It was presented that *Blastocystis* subtype ST-7 could stimulate programmed cell death in both an early and late stage. Additionally, the study found that *Blastocystis* ST-7 can activate the intrinsic pathway of apoptosis by the activation of caspase-3 and -9. In the case of caspase-8, no changes were identified. As for *Blastocystis* ST-4, no activation of the three caspases tested in this experiment was found [55].

The aforementioned studies suggest that *Blastocystis* is able to induce apoptosis in host cells. Most definitely, understanding all the mechanisms occurring during infection with this parasite will prove to be helpful in developing tailored therapeutic agents to prevent the negative interaction with host cells during blastocytosis, especially considering its asymptomatic course.

### 3.2. Giardia sp.

*G. intestinalis* (syn. *G. lamblia*, *G. duodenalis*) is a cosmopolitan protozoan belonging to flagellates. It is found in tropical and subtropical climate zones more often than in temperate zones. The parasite colonizes people, domestic animals (cats, dogs), and wild animals (beavers). According to the most recent literature on the subject, eight genotypes (A–H) of *G. intestinalis* have been identified with no morphological differences between them, being only different on the molecular level. Human giardiasis is caused by genotypes A and B, occupying mostly the small intestine and duodenum, however, in chronic giardiasis, *G. intestinalis* can also occupy the liver bile duct and the excretory ducts of the pancreas [56,57].

*G. intestinalis* occurs in two morphological forms: trophozoite and cyst (Figure 5). The adhesive disc of the *G. intestinalis* trophozoite acts as a type of sucker, allowing adhesion to the intestinal microvilli. The invasive form is the oval cyst, which contains two to four nuclei located closely together on one side of the pole.

Infection with this protozoan occurs via the fecal–oral route and sexual contact. In the case of the invasive form, the cyst enters the human body with water and food. The moment it enters the duodenum, the excystation process occurs and two trophozoites are released, which by means of adhesive discs attach themselves to the membrane of the small intestine. In some cases of giardiasis (during diarrhea), cysts and trophozoites are excreted [57,58]. Given the high invasiveness (<10 cysts) and resistance to external factors (water chlorination, low temperature), children are especially at risk of being infected with *G. intestinalis*. This particularly concerns small children attending nursery and kindergarten. *G. intestinalis* infection is diagnosed more frequently in undernourished and immunologically deficient patients (e.g., hypoglobulinemia IgA) [57].

The course of giardiasis is varied: asymptomatic, acute or subacute, non-inflammatory diarrhea, and chronic infection with undernourishment and weight loss. The most common symptoms of an infection include diarrhea, vomiting, flatulence, abdominal pain, nausea, and weight loss. Less frequent symptoms are a rash, food allergy, pruritus, and inflammation. In the chronic form of giardiasis, shortening of the intestinal villa is possible, which consequently leads to disturbed absorption, vitamin (A, B12, and folic acid), and lactase deficiency.

Giardiasis in people can be diagnosed with coproscopic tests such as direct smear, flotation, decanting (identification of cysts in feces, at times of trophozoites in diarrheal feces) and immunological tests: immunoenzymatic ELISA method (detection of coproantigen *G. intestinalis*) by means of the direct immunofluorescence test (detection of cyst wall antigens *G. intestinalis*). *G. intestinalis* can also be identified using molecular biology methods, as used for scientific purposes, and the obtained results determine not only the type of parasite, but also its genotype [57,58].

Recent studies demonstrate that *Giardia* sp. can induce apoptosis in intestinal epithelial cells by activating both the intrinsic and extrinsic pathways of apoptosis [59]. Numerous studies suggest that depending on the species, the parasite may have a destructive effect on the intestinal epithelium of the host through the induction of various mechanisms. It can stimulate apoptosis by activating caspase-3, changes in protein expression including high expression of proapoptotic proteins (Bax) and simultaneous low expression of antiapoptotic proteins (Bcl-2) as well as PARP inactivation [60,61,62,63]. A study by Chin et al. (2002) found that trophozoites of *G. intestinalis* can activate apoptosis in non-transformed human epithelial cells of the small intestine, depending on the strain of the parasite. Moreover, analysis of the results shows that in inducing apoptosis, *G. intestinalis* also affects the permeability of the intestinal epithelium barrier of the host, leading to its loss. Additionally, it was demonstrated that these changes may be dependent on caspase-3 [61]. The ability of various strains of *G. intestinalis* to cause changes in the cells of the small intestine epithelium was also confirmed by Cevallos et al. (1995), who conducted a study on neonatal Sprague–Dawley rats. It was found that isolates of *G. intestinalis* produced different degrees of perturbation of mucosal structure or function. Villus height in infected rats was reduced without evidence of crypt hyperplasia. Furthermore, infected rats had a severe impairment of absorption of water and electrolytes with a morphologically intact mucosa [64]. Another study on apoptosis in the parasite–host system was conducted by Panaro et al. (2007) using human ileal adenoma cell line (HTC-8). The interaction of the epithelial cells of the colon with *G. intestinalis* trophozoites showed that this parasite can modulate the processes of apoptosis in the intestines of the host by DNA fragmentation, stimulation of caspase-3, PARP degradation, and regulation of protein expression Bcl-2 and Bax. It was found that the developed adaptive mechanisms of the parasite, consisting of surface protein production and secretory products, may be responsible for inducing apoptotic pathways. The results clearly indicate the proapoptotic potential of *G. intestinalis* and suggest the function of caspase-dependent apoptosis in the pathogenesis of lambliosis [59]. An experiment conducted by Troeger et al. (2007) demonstrated only a slight effect of *G. intestinalis* on apoptosis among patients infected with giardiasis. Presumably, apoptosis is triggered by direct contact of the protozoan or its products with the epithelium. Furthermore, the coefficient of apoptosis during intestinal giardiasis was found to have increased moderately to 0.2% in comparison to the control −0.1%. The results suggest that even a low-level release of proinflammatory cytokines such as TNF-α or IFN could lead to increased epithelial apoptosis in response to chronic giardiasis. The apoptosis of intestinal epithelial cells can play a significant role in the epithelial barrier function, even if the increase is only moderate [63]. Furthermore, Roxstrom-Lindquist et al. (2005) suggest that the said parasite not only has the ability to induce apoptosis, but also trigger major changes in the expression of the genes from tissue cells such as proinflammatory factors from the cells of the intestinal epithelium [62].

However, all experiments so far concerning the interaction between *Giardia* and the host are insufficient to precisely explain the mechanisms of apoptosis stimulation in the cells of the intestinal epithelium of the host as a result of infection with *Giardia* sp.

### 3.3. Cryptosporidium sp.

Over a dozen species within the *Cryptosporidium* genus have been identified, the most abundant being *C. parvum* (more than 90 types). Another species frequently infecting humans is *C. hominis*. Due to genotype diversity within this genus and the similar morphology of oocysts, it is difficult to diagnose the infection only by microscope. Only by means of molecular identification is it possible to differentiate the human parasites from the forms not causing infection in humans (Figure 6).

*Cryptosporidium* is a genus of obligate intracellular protozoan parasites. They mostly occupy epithelial cells of the small intestine, however, they are also found in the pancreas and bile duct. The invasive form of *Cryptosporidium* sp. is an oocyst characterized by a high resistance to water purification processes. The minimum infectious dose for a healthy individual is approximately 100–200 oocysts, while a dose of 30 cysts can produce symptoms in an immunocompromised individual, where cryptosporidiosis can result in cachexia and even death. Additionally, auto-invasion is often reported in such patients. Infection is mostly recorded in small children and oncology patients. The most characteristic symptom of the infection is a profuse watery diarrhea and abdominal pain. Additionally, nausea, vomiting, and fever can be reported [65,66,67].

Similar to the previously discussed intestinal parasites, *Crytosporidum* can contribute not only to a disturbance of individual cells, but also to disturbing the functioning of the entire organ, given that the parasites develop complex mechanisms that modulate the parasite–host interaction. Depending on the species and particular stages of the developmental cycle, it has been demonstrated that this parasite affects the mechanisms of the host apoptotic pathways [68,69,70,71,72,73,74]. Due to the enteropathogenic effect of *Cryptosporidium*, Buret et al. (2003) investigated the interaction of infection with *C. andersoni* with human and bovine epithelial cell lines with respect to apoptosis induction. The experiment consisted of a quantitative determination of nuclei in the epithelial cells using Hoechst staining. Using confocal microscopy, visible signs of fragmentation and condensation of chromatin, which are characteristic for apoptosis, were demonstrated. It was expressly stated that the exposure to *C. andersoni* induced a significant increase in apoptotic indices (1.8 ± 0.27%) compared to the control (0.23 ± 0.08%). Additionally, the application of epidermal growth factor (EGF) to the cell culture resulted in the inhibition of apoptotic processes induced by *Cryptosporidium* [68]. Furthermore, it was found that the process is extremely complex and determined by numerous factors, among others, by the research model, duration of infection, and the developmental stage of the parasite [69,70,71,72,73,74,75,76,77]. A study on the effect of HCT-8 interaction with *C. parvum* in different developmental stages conducted by Mele et al. (2004) demonstrated significant differences in the induction of apoptotic pathways. The experiment allowed identification of the inhibitory apoptotic potential in the trophozoite stage of *Cryptosporidium,* and the potential, which induces apoptosis in the sporozoite and merozoite stage [71]. Additionally, Chen et al. (1999) observed that cryptosporidiosis may induce apoptosis in a human biliary epithelium cell culture. It was found that the parasite influences apoptosis in the biliary epithelium by a mechanism dependent on Fas/ FasL, which includes both autocrine as well as paracrine cellular signaling [69]. McCole et al. (2000) stated that *C. parvum* could have a contradictory effect on apoptosis levels in human intestinal epithelial cell cultures. The parasite can induce a moderate level of apoptosis in the structure under analysis, and on the other hand, it can show properties inhibiting apoptosis as a response to strong proapoptotic stimuli. It may be that inhibition of apoptosis is due to the growth and development of the parasite. Induction in the early stages of moderate apoptosis levels inhibits the occurrence of inflammation in the host and, at the same time, maintains the integrity of the epithelial barrier. The results by McCole et al. (2000) suggest the development of an adaptive mechanism by the parasite in order to complete its developmental stage [70]. Similar results were obtained by Liu at al. (2009) using micro-array and human ileal adenoma cell model HCT-8 to analyze the two-phase modulation of apoptotic mechanisms during cryptosporidiosis. The large scope of the study showed the multifaceted character of the parasite–host interaction, and also identified the panel of proteins connected with apoptosis, the genes of which altered expression during *C. parvum* infection. The meta-analysis of the profile of apoptotic genes transcription suggests that the balance of gene expression in the early phase of infection, with respect to antiapoptotic genes, was induced; whereas with respect to proapoptotic genes, the balance was inhibited. The opposite was observed in the late stage of infection when proapoptotic genes were induced and antiapoptotic genes were inhibited. Additionally, it was found that the apoptotic pathway limits the development of the parasite [75]. The studies related to the early phase of *C. parvum* infection in an interaction with HCT-8 cells were also conducted by Wang et al. (2019). Studies on the microRNA expression profile demonstrated the inductive potential of the parasite with respect to apoptosis. The molecular mechanisms triggered by *C. parvum* infection point to an interdependency between the parasite and the host at the miRNA level [77].

Programmed cell death is a crucial process for both the survival of the parasite as well as for its pathogenicity, where the apoptotic response of the infected host cell is actively suppressed by the parasite via upregulation of survivin, favoring the parasite in order to complete its developmental stage. *Cryptosporidium* may have evolved means to control host intestinal epithelial cell apoptosis to facilitate the establishment of infection in the newly exposed host. Furthermore, it can be concluded that infection with this parasite is related to the immune response, and adaptation of the host organism to the antigens of this parasite. Changes were observed in intestinal epithelial cells in a rat model, indicating a defense against parasitic infection. Overall, information on the pathomechanisms underlying gastrointestinal disorders such as intestinal disorders during cryptosporidiosis are not comprehensive and need further research. Furthermore, due to difficulties in preventing cryptosporidiosis and the fact that *Cryptosporidium* spreads easily, studies on the parasite–host system with respect to modulation of the apoptotic pathways seem to be an invaluable source of data for the diagnostics and treatment of this particular parasitosis.

### 3.4. Trichuris sp.

Studies on the parasite–host interaction, also with respect to *Trichuris sp*., have been carried out for many years. This particular parasite is widespread both in humans and mammals (ruminants, marsupials, rodents). Infection results from the ingestion of eggs present in contaminated food, water, or soil. Once the eggs are swallowed, the hatched larvae migrate through the gastrointestinal tract, and develop in the cecum until reaching maturity. Adult specimens enter the mucus membrane of the cecum, less frequently of the large intestine. Adult females of *Trichuris sp*. lay up to 5000 eggs daily, which are then excreted by the host in the feces. Following 20 or more days in a moist environment with oxygen, the eggs ultimately mature, reaching the invasive phase (Figure 7). The eggs of *Trichuris* sp. are resistant to both drying out and extreme temperatures. Due to the capability of infecting different species of hosts, *Trichuris* sp. is considered to be a parasite of epidemiological concern. As a result of difficulties associated with the high reproductive potential and widespread occurrence, the said parasite has the capability to reappear in a location from which it has been previously eliminated. The most common epidemiological factor for humans is *T. trichura*. This nematode causes a parasitic disease called trichuriasis, also known as whipworm infection. In humans, trichuriasis usually has a mild course, producing only mild symptoms or is asymptomatic. At times, *T. trichura* may cause severe infections of the gastrointestinal tract including chronic diarrhea with loose bloody mucus stool, abdominal pain, and anemia. In the case of a massive infection, there may also be intestinal ulceration corresponding to the clinical image and resembling ulcerative colitis. The frequency of trichuriasis in humans is high. It is estimated that 600–800 million people are infected with *T. trichura* worldwide. The diagnostic process mostly relies on determining the presence of *T. trichura* eggs in stool samples, however, the similarity of eggs of different species of *Trichuris* sp. makes the process difficult [78,79,80].

Due to the parasite’s habitat and the effect it has on the functioning of the intestines, studies on the modulation of apoptotic pathways in the intestines in the parasite–host system are of particular interest. It is well-known that *Trichuris* sp. triggers changes in the mucosal barrier during acute and chronic infections. Investigating the mechanism will surely contribute to establishing better treatment options in whipworm infection [81]. The study model used most frequently is *T. muris* species, a natural intestinal mouse parasite, which serves as an equivalent to *T. trichura* occurring in humans. The results by Cliffe et al. (2007) suggest that apoptosis can be induced due to chronic infection with *T. muris* in the large intestine of the infected mice, and not because of the physical damage caused by the parasite. Moreover, it was found that the secretion of proinflammatory cytokines TNF-α and IFN-γ in the course of trichuriasis plays a role in programmed death of the cells of the intestinal epithelium. These studies have proven that the effect of infection with *Trichuris* species is not necessarily connected with just the number of parasites [82]. The previous results by Cliffe et al. (2005) suggest that *T. muris* is a parasite increasing the rate of cell turnover in the host’s large intestine in order to excrete the parasite. The study demonstrated that the phenomenon is controlled by the immune system of the host, depending on the secretion of cytokines:interleukin-13 (IL-13), increasing the cell turnover rate of the epithelium and removing the parasite. The mechanical stimulation of the host’s defensive mechanism is negated by the pathogen through its effect on the secretion of chemokine CXCL10, which decreases the epithelial cell turnover rate in a mouse model of colitis and causes cell hyperplasia in crypts. Controlling *Trichuris* sp. infection on the surface of the mucus membrane of the large intestine is determined by a series of defense mechanisms on the part of the host’s immune system [83]. Additionally, the transcriptional profiling of the large intestine of a mouse infected with *T. muris* shows that the parasite stimulates the immune system of the host by affecting numerous molecular pathways such as apoptosis. An analysis of the gene expression using micro-array allowed for the determination of the course of inflammatory bowel disease (IBD) in mice, and indicated a high degree of similarity with the course of this disease in humans [84]. Another study revealed the predominant role of protein p55 for TNF-α receptor in *T. muris* infection. As mentioned earlier, apoptosis is induced by means of TNF-α via p53, which directly attaches to the death domain in the apoptotic pathway. Furthermore, the studies confirm significant differences in the excretion of the parasite with respect to the sex of the host. It was found that the female mice could eliminate *T. muris* up to day 35 of the infection, whereas the male mice exhibited a chronic course of the infection [85]. The most recent study by Hayes et al. (2017) indicated that both cell proliferation and apoptosis increased in the large intestine during chronic *T. muris* infection in mice. At the base of the crypts, a marked increase in the number of cells indicative of apoptosis was observed, and in more severely infected animals, there were successively more proliferating cells in the further parts of the intestinal crypts. Interestingly, there was no effect of the infection on proliferation and apoptosis in the small intestine, despite the occurrence of morphological changes at this site [86].

Therefore, it seems that the sex of the host is relevant with respect to the induction of the apoptotic pathway in *Trichuris* sp. infection. Considering the effects of whipworm infection, it only seems reasonable to conduct further studies aimed at a better understanding of the said dependence. Ongoing research is focused on determining whether disorders caused by *Trichuris* sp. in the cycle of intestinal epithelial cells affect the induction of apoptosis. The validity of research on *Trichuris* sp. is confirmed by the numerous morphological limitations encountered nowadays when attempting to diagnose the parasite infection. Moreover, there is a common phenomenon of morphological convergence in nematodes, therefore the use of molecular markers for the purpose of identifying the species of a parasite such as *Trichuris* sp. is stressed. Currently available literature data regarding the interaction between *Trichuris* sp. and the host with respect to the stimulation of apoptosis is lacking, and further research is needed.

### 3.5. Entamoeba Histolytica

*E. histolytica* is a cosmopolitan unicellular parasite of the large intestine. In humans, it is present in two forms: intestinal, manifested by bloody mucous diarrhea, flatulence, abdominal pain, weight loss, elevated temperature; and extraintestinal, characterized by the presence of abscesses in, for example, the liver and lungs. This parasite is considered dangerous to humans and causes a condition called amoebiasis (amoebic dysentery). As untreated amoebiasis shows a high mortality, particularly by the intestinal form, studies on this infectionare necessary. Since the gastrointestinal tract of a human can be colonized by other pathogenic species from the *Entamoebidae* family, which are defined as belonging to commensals *E. dispar*, *E. coli*, *E. hartmanni*, *E. polecki*, a differential diagnosis with pathogenic *E. histolytica* is crucial. *E. histolytica* colonizes an organism in two morphological forms: an encapsulated cyst as the dormant form, and a trophozoite as the vegetative form (Figure 8). Proteolytic enzymes secreted by the trophozoite form (hyaluronidase, cysteine protease) can lead to cytolysis and cell degradation. To continue its developmental cycle, the parasite requires specific conditions (moisture, temperature) as well as a host. The optimum conditions for *E. histolytica* are in tropical and subtropical climate zones. It was shown that *E. histolytica* is prevalent in the Indian Peninsula, South America, Bangladesh, the Republic of South Africa, and in Italy. Infection takes place via ingestion of water or food contaminated with cysts, at times through contamination of the oral cavity due to poor hygiene habits while performing household chores as well as via homosexual activities among men [87,88,89].

Given their ability to cause lysis of the host cell, research on the effect of virulence factors of *E. histolytica* and modulation of apoptotic pathways due to infection is essential. The mechanisms of *E. histolytica* cytotoxicity are still being discussed. It was proven that the parasite can cause cell cytolysis through direct contact. Death of the target cells is associated with caspase-dependent activation on contact with the infected cells [90]. Additionally, it was found that the virulence and *E. histolytica* trophozoite adherence to the cells of the intestinal mucosa is connected with lectin receptor Gal/GalNAc. Thus, the extracellular parasite can induce apoptosis in cells, irrespective of the membranous death receptors (i.e., an extrinsic apoptotic pathway). This parasite has the ability to induce apoptosis through direct contact when binding galactosamine to surface lectin. This allows a subsequent spreading of the parasite with blood through the portal vein system to distant sites such as the peritoneum, liver, lung, or brain [87]. It was demonstrated that the cytotoxicity of *E. histolytica* also depends on effector proteins, which include amoeba pores and proteases responsible for the parasite invasion and apoptosis of the host’s cell. Moreover, it is believed that these proteins are responsible for the direct activation of caspase-3 [91]. Similar results were obtained by Huston et al. (2001) in infected mice –C3H/HeJ *E. histolytica*. Studies on the tissue of the large intestine of the infected mice showed a rapid induced activation of caspase-3, independent of caspase-8 and -9 [92]. Earlier studies on the relationship between *E. histolytica* infection and apoptosis also showed a strong induction of host cell death due to the presence of this parasite. The study by Ragland et al. (1994) [93] confirmed this hypothesis. The results showed a characteristic pattern of DNA degradation in the mouse myeloid cell line FDC-P1. Additionally, the activation of apoptosis was independent of the over expression of the Bcl-2 apoptotic mechanism [93]. Similarly, the experiment by Seydel et al. (1998) [94] confirmed the induction of apoptosis in hepatocytes in a mouse model of a severe combined immunodeficiency (SCID) during infection with *E. histolytica* trophozoites. The study indicated that the process takes place independently of the activation of the Fas/Fas ligand pathway of apoptosis, without mediation of TNF-α [94]. Studies on the tissue of the large intestine of mice point to the role of apoptosis in infection with *E. histolytica*. The strategy adopted by the parasite most likely is a new adaptive trait and facilitates its virulence [95].

### 3.6. Nippostrongylus Brasiliensis

*N. brasiliensis* is an intestinal nematode of rodents (mouse, hamster, rabbit, chinchilla, and rat). This parasite does not occur naturally in laboratory rodents. Due to close affinity with human nematode (*Necator americanus*, *Ancylostoma duodenale*) and a simple life cycle, *N. brasiliensis* is a particularly useful model for the study of infection with intestinal parasites in humans, especially with respect to the induced immune response of the host. In experimental studies, for the purpose of the observation of the life cycle and parasite–host interaction, laboratory mice are used most often. The availability of inbred and mutated strains of mice can be used for genetic studies of the susceptibility to *N. brasiliensis* infection. The use of such an animal model can help in better understanding the induction and the course of the humoral immune response dependent on the subpopulation of the helper cells Th2. It was manifested that following infection with *N. brasiliensis*, laboratory animals develop massive emphysema and severe obstructive pulmonary disease (SOPD). Moreover, morphological changes in the lungs are observed including damage to the alveoli and long-term airway hypersensitivity. Apart from respiratory problems, the infection may cause weight loss and a decreased erythrocyte levels. The initial phase is associated with parasites invading the lungs, whereas the following phase occurs in the intestines where the parasite reaches maturity [96,97,98,99,100].

In parasitic infections, changes in intestinal morphology are observed including atrophy of villi. Similar findings have been reported in *N. brasiliensis* infection. The studies by Hyoh et al. (1999) proved that this parasite can cause a partial atrophy of villi and crypt hyperplasia in the mucosa of the small intestine. Moreover, it was found that villi atrophy was connected with increased apoptosis and the loss of adhesion in the epithelial cells of the host [101]. Thus, induction of the apoptotic pathway in the intestinal epithelial cells can be associated with the defense mechanism on the part of the host employed to quickly remove the damaged cells. It is suspected that apoptosis of the cells of the villi may be directly caused by the particles originating from the nematode itself: excretory–secretory products (ES). It was found that ES of *N. brasiliensis* contains biologically active particles such as acetylcholinesterase, proteases, and a factor inhibiting the production of interferon gamma [102,103]. The experiment conducted by Kuroda et al. (2002) investigated the effect of worm extract (WE) and ES of *N. brasiliensis* on the cell line of the intestinal epithelium IEC-6. These studies explicitly indicate that *N. brasiliensis* secretes biologically active particles that are able to induce apoptosis in intestinal epithelial cells of the host with an increase in Fas expression. Even though the mechanism of the induction still needs further research, it was proven that exposure to ES and WE triggered apoptotic mechanisms in the cells including nucleus fragmentation, activation of caspase-3, and specific PARP fission [104]. The studies conducted so far provide valuable data on the stimulation of apoptosis of the host’s cells alone, however, they do not fully explain the mechanisms underlying the induction. Certainly, further experiments may contribute to a better understanding of the issue.

### 3.7. Heligmosomoides Polygyrus

*Heligmosomoides polygyrus* (formerly *Nematospiroides dubius*) was first isolated from wild California mice in the 1940s. Since then, the parasite has been used in laboratories worldwide as the model parasite in studies on parasite–host interaction. As in the case of *N. brasilienis*, the course of *H. polygyrus* infection ideally represents a parasitic infection of the gastrointestinal tract in humans and domestic animals.

It was proven that this natural intestinal parasite of mice has a strong immunomodulatory effect on the immune system of the hostnot only through the mere presence of the parasite, but also through its secretory products, HES (*H. polygyrus* excretory-secretory antigens) [105,106]. The studies demonstrated that the larval stages of *H. polygyrus* result in a strong response dependent on Th2 lymphocytes in the primary phase of infection. In the chronic form of the infection, adult parasites stimulate the regulatory reactions, leading to a reduced immune response of the host’s lymphocytes. The relationship between nematode-induced immunosuppression is reflected by the hypo-responsiveness as well as by the inhibition of apoptosis. Additionally, it is suspected that the parasite’s antigens, which prevent glucocorticoid-induced apoptosis, affect the number of T regulatory cells (Treg) and apoptosis of both T CD4 as well as CD8-positive cells. The observations suggest that the *H. polygyrus* proteome contains immunomodulatory factors responsible for the avoidance of the immune response in the host [107,108,109]. The studies by Donskow-Schmelter et al. (2007), which compared proliferation, apoptosis, and production of IL-2 and IL-6 MLN (mesenteric lymph node) and spleen cells in vitro from fast responder FVB mice and slow responder mice (C57Bl/6) infected with *H. polygyrus*, demonstrated an important role in the modulation of the host’s response during infection. It was proven that during infection, secreted IL-2 and IL-6 play a potential role in apoptosis and T cell survival through increased transcription and translation of the FasL gene. Programmed cell death by FasL protein is of significant importance to the survival of T cells in vitro, which suggests a correlation between IL-2 and IL-6 synthesis, apoptosis, and a proliferative response during *H. polygyrus* infection [108]. Furthermore, the results by Donskow-Łysoniewska et al. (2013) confirm that exposure to *H. polygyrus* antigens has an inhibitory effect on the intrinsic pathway of apoptosis by the over expression of survivin and antiapoptotic protein Bcl-2 in T CD4 cells. Additionally, it was demonstrated that proliferation induced by fraction 9 (F9Ag) of the somatic antigens of the nematode was dependent on a low Bax/Bcl-2 ratio, and that the inhibition of apoptosis is dependent on caspase-3 and is independent of caspase-8 [110].

Proteomic studies on the complexity of *H. polygyrus* antigen composition will surely contribute to determining the significance of the parasite’s particles in fast as well as slow responder hosts during infection.

## 4. Conclusions

The information provided in the present paper proves that apoptosis constitutes only one of the many mechanisms occurring during infection with intestinal parasites, and that the determined parameters can serve as useful markers of apoptosis in parasitic diseases. Many questions have not yet been answered such as whether induction of the physiological epithelial cell turnover is stimulated at the immunological level to remove the parasites, at the same time causing their final excretion along with their direct environment, the intestinal epithelium; in what way inhibition of apoptotic pathways affects the host’s intestinal epithelium; whether it is possible to administer antiapoptotic treatment during parasitic infections without causing harm to other tissues. In light of the above-discussed information on the modulation of apoptosis in the host during a parasitic infection, it can be inferred that the issue is not fully understood and still requires further studies using other intestinal parasites. Given the wide scope of the issue and the increasing number of infections with “new” and opportunistic intestinal parasites, the prospects for further studies are indeed interesting. Furthermore, due to the lack of explicit and widely available animal models used for studying infections with intestinal parasites, further experiments should also focus on developing the animal model to further widen the scope of in vitro studies of the parasite–host system.

## Figures and Tables

**Figure 1 biomolecules-10-00674-f001:**
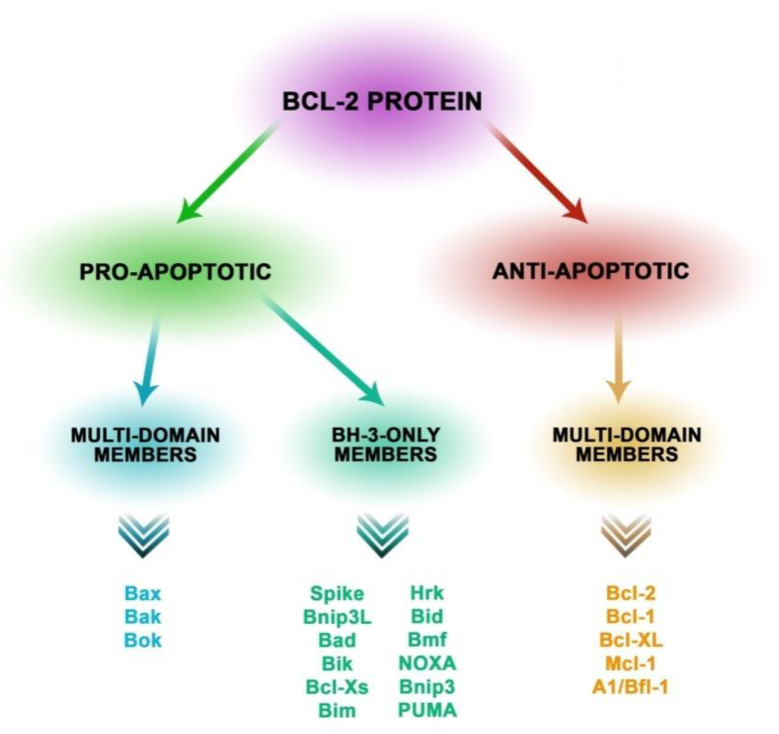
Classification of proteins from the Bcl-2 family into three subfamilies determined by the presence of the Bcl-2 domain. The family of proapoptotic Bax proteins structurally contains the BF1-3 domain and trans membrane domain (TM). This genus of subfamily forms channels in the mitochondrial membrane. The representatives are Bax, Bak, and Bok. Proapoptotic proteins also include the subfamily with the BH-3-only domain such as Bad, Bim, Bik, Bmf, Bid, Bnip3L, Bnip3, Hrk, Noxa, Puma, and Spike. The essential function of these proteins is to induce apoptosis as a response to cellular stress. The subfamily of apoptotic proteins that structurally contain BH 1-4 and TM has the function of maintaining the integrity of the mitochondrial membrane. This subfamily includes the following proteins: Bcl-2, Mcl-1, Bcl-XL, Bcl-w, Bcl-1, and A1 [20,21,22,23,24,25].

**Figure 2 biomolecules-10-00674-f002:**
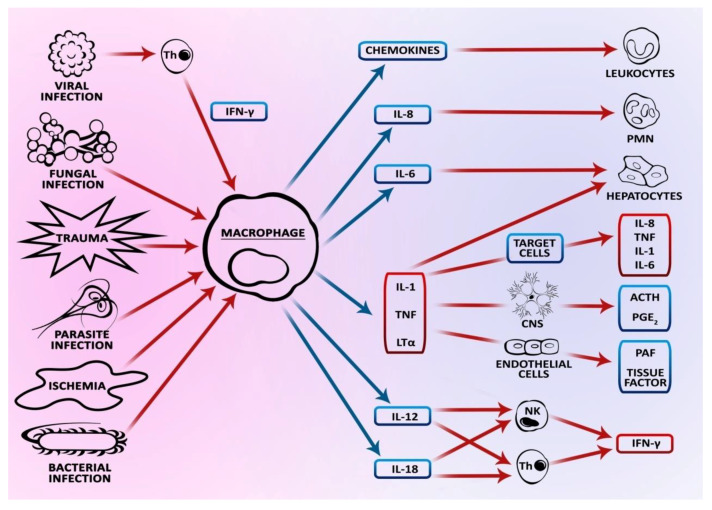
Relationships between cytokines.

**Figure 3 biomolecules-10-00674-f003:**
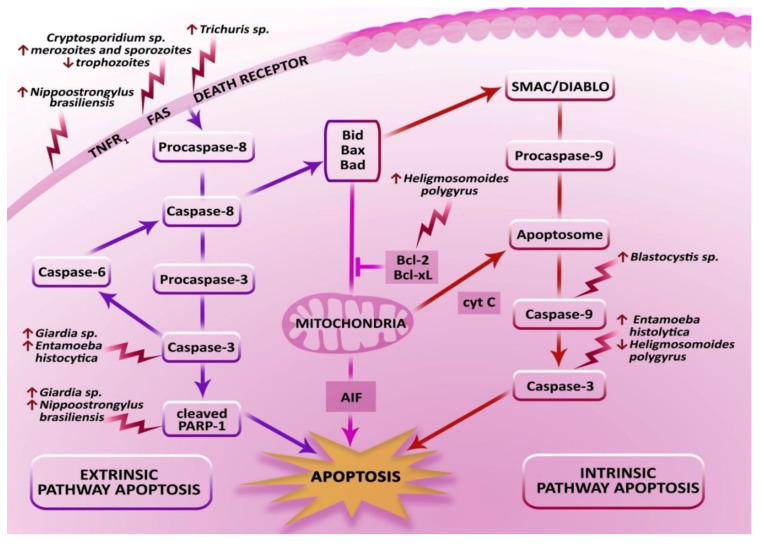
Induction of apoptotic signals in enterocytes by selected intestinal parasites.

**Figure 4 biomolecules-10-00674-f004:**
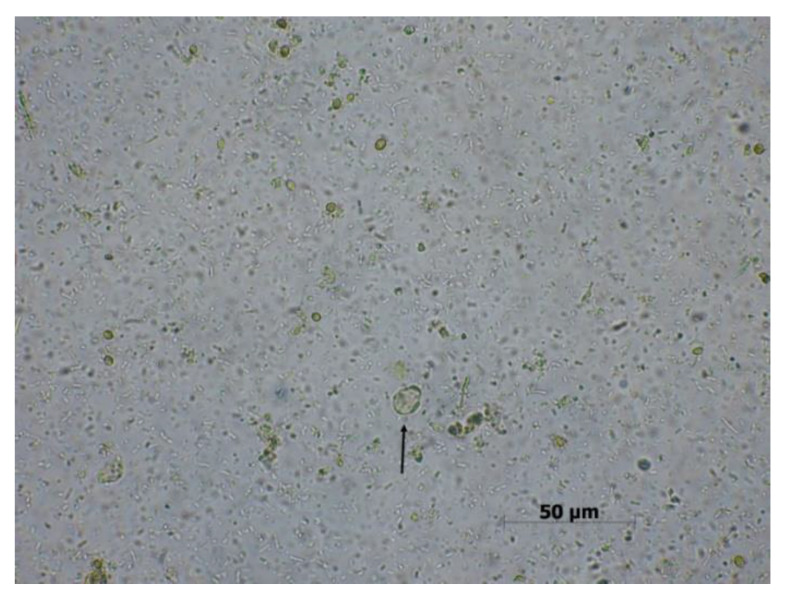
*Blastocystis hominis* vacuolar forms found in a wet mount sample at 40x magnitude (original photography).

**Figure 5 biomolecules-10-00674-f005:**
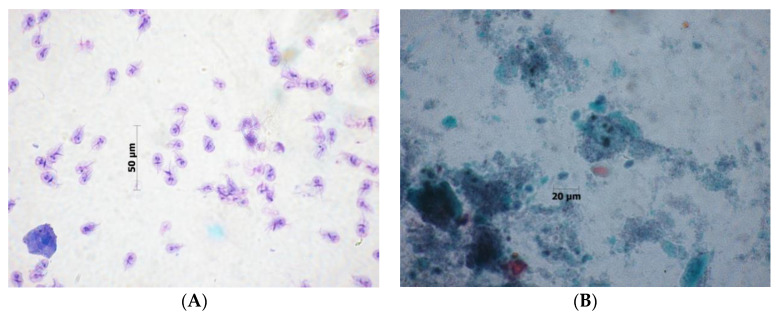
(**A**) *Giardia* sp. trophozoite forms at 40x. (**B**) *Giardia* sp. cyst forms at 40x (all original photography).

**Figure 6 biomolecules-10-00674-f006:**
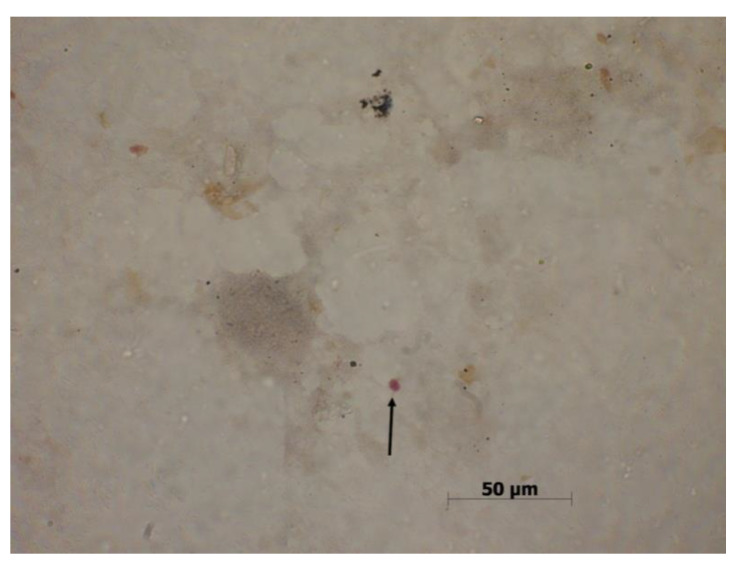
*Cryptosporidium* sp. at 40x (original photography).

**Figure 7 biomolecules-10-00674-f007:**
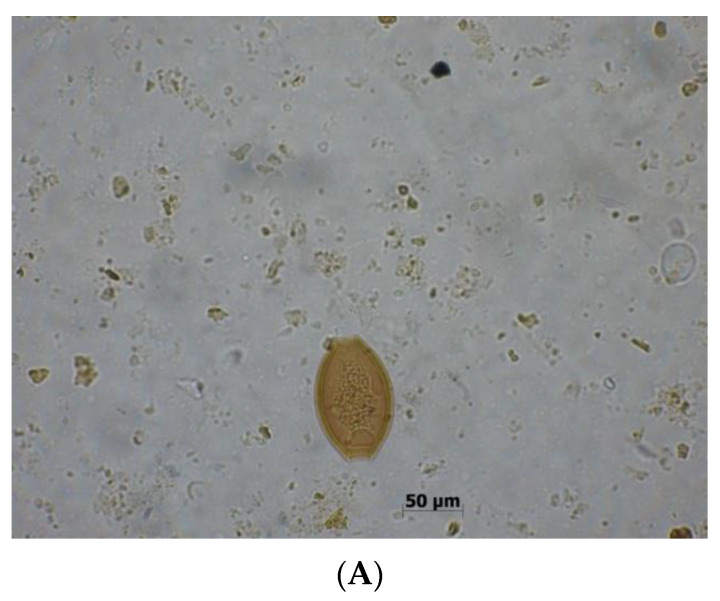
(**A**) Egg of *T. trichiura* in an iodine-stained wet mount at 40x. (**B**) Adult of *T. trichiura* at 4x (**C**) Adult of *T. trichiura* at 4x (all original photography).

**Figure 8 biomolecules-10-00674-f008:**
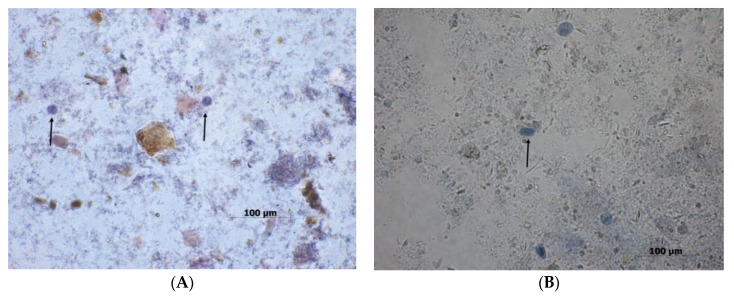
(**A**) Cyst of *E. histolytica* at 40x. (**B**) Trophozoite of *E. histolytica* at 40x (all original photography).

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
