# Peer review of "The Influence of Selected Gastrointestinal Parasites on Apoptosis in Intestinal Epithelial Cells"

_biomolecules, 2020, doi:10.3390/biom10050674_

Round 1

Reviewer 1 Report

Decision: Major revision

In the manuscript by Kapczuk et al., the authors summarize the interaction between gastrointestinal parasites and host intestinal epithelial cells from the view of apoptosis regulation. However, as the authors themselves describe in the Conclusions section, “apoptosis constitutes only one of the many mechanisms occurring during infection with intestinal parasites”. The authors may intentionally exclude the description of T-cell apoptosis, cytokine-mediated immune response, or inflammation process, but this makes the readers difficult to image an integrated whole picture of host-parasite interaction. In addition, the authors selected seven parasite species, and readers may be interested in the appearance, life style, and molecular mechanisms of apoptosis induction. The information should be visually provided by photographs and illustrations. Moreover, the second paragraph is devoted to the general molecular mechanism of apoptosis, but contains several wrong or inaccurate descriptions. Also, several technical terms are incorrectly used. Therefore, I would like to recommend this review manuscript for the publication in Biomolcules after the consideration of following points.

MAJOR POINTS

In Figure 1, the members of pro-apoptotic BCl-2 family proteins are wrong, although figure legend is correctly described. Bax, Bak and Bok are multi-domain members, and the other proteins including Bid and Bim are BH3-only members. Moreover, there is no protein known as “Beklina1”. Although it may be “Beclin 1”, it has been widely accepted that beclin 1 is directly involved in autophagy rather than apoptosis. Also, Bik and Blk (and Nbk) are the same protein.

The readers may be interested in life style, and molecular mechanisms of apoptosis induction of the parasites selected by the authores. Please provide the visual information of these parasites with either photographs or illustrations.

It will be necessary to include a figure summarizing the function of immune cells and inflammatory cytokines. This will help the readers to understand the physiological function of TNFαor IFN, which frequently appear in the manuscript.

In lines 42 and 199, please describe the meaning of “particles”. Do they mean specific higher-order molecular structures? From the context, I personally feel that “particle” may be “protein”.

In Line 125, TRAIL is a ligand but not a receptor. DR4 and DR5 are currently known TRAIL receptors.

MINOR POINTS

Throughout the manuscript, please unify the hyphenation rule of caspase family members.

In Figure 2, “Apoptosom” should be “Apoptosome”, and “INTERINSIC” should be “INTRINSIC”. Also, what is the meaning of “DAS”?

In lines 111 and 115, please describe the meanings of “TAD”, “TAD2”, and “AD2”. Are they transactivation domain?

In line 132, please use correct terms. What is “kinase I”? Is it RIP1? NF-kB is “nuclear factor kappa B” and does not require “factor”. JNK and p38 are both protein kinases.

In Line 142, “external mitochondrial membrane” should be “mitochondrial outer membrane”.

In Line 151, “Bcl” should be Bcl-2, Bcl-XL or so.

In Line 164, “Omni” should be “Omi” as designated in Line 155.

In Line 380, “EFG” should be “EGF”.

In Lines 394-395, “On the other hand” appears redundantly.

Author Response

Reviewer 1

Authors: Thank you very much for the in-depth evaluation of the article, a positive opinion, as well as the critical remarks which provide important indications that helped us improve the quality of the article.

Reviewer 1: In Figure 1, the members of pro-apoptotic BCl-2 family proteins are wrong, although figure legend is correctly described. Bax, Bak and Bok are multi-domain members, and the other proteins including Bid and Bim are BH3-only members. Moreover, there is no protein known as “Beklina1”. Although it may be “Beclin 1”, it has been widely accepted that beclin 1 is directly involved in autophagy rather than apoptosis. Also, Bik and Blk (and Nbk) are the same protein.

Authors: Thank you for indicating the additional points. They are now included in Figure 1. This is our mistake.

Reviewer 1: The readers may be interested in life style, and molecular mechanisms of apoptosis induction of the parasites selected by the authores. Please provide the visual information of these parasites with either photographs or illustrations.

Authors: We have supplemented the article with microscopic images of parasites mentioned in the text

Reviewer 1: It will be necessary to include a figure summarizing the function of immune cells and inflammatory cytokines. This will help the readers to understand the physiological function of TNFαor IFN, which frequently appear in the manuscript.

Authors: We added of figure summarizing the function of immune cells and inflammatory cytokines.

Reviewer 1: In lines 42 and 199, please describe the meaning of “particles”. Do they mean specific higher-order molecular structures? From the context, I personally feel that “particle” may be “protein”.

Authors: Corrected according to the indications of Reviewer. We changed to ,,antigen protein” in line 42 and ,,level” in the line 199.

Reviewer 1: In Line 125, TRAIL is a ligand but not a receptor. DR4 and DR5 are currently known TRAIL receptors.

Authors: We changed to ,, The extrinsic pathway is determined by the interaction of ligands with membrane receptors belonging to the superfamily of tumour necrosis factor (TNF) receptors: TNFR1, TNFR2, Fas/CD95/Apo1 and ligands TRAIL (TNF- related apoptosis-inducing ligand)/Apo2.” in the line 125.

Reviewer 1: Throughout the manuscript, please unify the hyphenation rule of caspase family members.

Authors: Thank you very much for the suggestions. Corrected according to the indications of Reviewer.

Reviewer 1: In Figure 2, “Apoptosom” should be “Apoptosome”, and “INTERINSIC” should be “INTRINSIC”. Also, what is the meaning of “DAS”?

Authors: Thank you for indicating the additional points. They are now included in Figure 2. This is our mistake.

Reviewer 1: In lines 111 and 115, please describe the meanings of “TAD”, “TAD2”, and “AD2”. Are they transactivation domain?

Authors: The missing information is now provided in the text.

Reviewer 1: In line 132, please use correct terms. What is “kinase I”? Is it RIP1? NF-kB is “nuclear factor kappa B” and does not require “factor”. JNK and p38 are both protein kinases.

Authors: Supplemented by ,, As a result of formation of complex I, the following are activated: receptor-interacting serine/threonine-protein kinase 1 (RIPK1), nuclear factor kappa B (NF-KB) and protein kinases: c-Jun N-terminal kinases (JNK) and p38 protein”

Reviewer 1: In Line 142, “external mitochondrial membrane” should be “mitochondrial outer membrane”.

Authors: Corrected.

Reviewer 1: In Line 151, “Bcl” should be Bcl-2, Bcl-XL or so. In Line 164, “Omni” should be “Omi” as designated in Line 155. In Line 380, “EFG” should be “EGF”.

Authors: Corrected.

Reviewer 1: In Lines 394-395, “On the other hand” appears redundantly.

Authors: The fragment “On the other hand” has been delete in lines 394.

Reviewer 2 Report

This is an interesting and informative review. The English is clumsy in places but not wrong. There are a few errors in the English and I have corrected these.

Author Response

Reviewer 2

This is an interesting and informative review. The English is clumsy in places but not wrong. There are a few errors in the English and I have corrected these.

Authors: Thank you very much for the in-depth evaluation of the article and positive opinion.

Reviewer 3 Report

Whilst looking forward to reading this paper, I was ultimately somewhat disappointed.  Partly this was down to a poor quality of english, but also since I felt an opportunity was missed.  As a review article, there was the chance to provide an overview on why parasites might seek to promote, or to prevent apoptosis of epithelial cells, presumably since either might favour the biology and longevity of infections with the various organisms, or even to describe why control of apoptosis might be advantageous from the host's perspective, but such an overview was largely lacking.  Instead the review rather reads as a list of studies and their findings.  As an example, if villous atrophy can be mediated by substances released by N. brasiliensis that promote apoptosis, why would the parasite choose to do this when its presence in the gut is largely determined by its ability to coil its body around the villi, so that if they are lost, the parasite loses its ability to maintain its position in the gut.  Such processes perhaps speak more of the host response to infection than to mechanisms the parasites use themselves to survive.  It also illustrates the difficulty between determining the differences that occur between disease states associated with infection, and infections with the same parasite in the same host where disease does not arise.  The latter situations arguably being more relevant to the natural host parasite relationship.

The review would also have benefited from a clearer delineation of evidence from in vivo studies versus in vitro studies.  Since the latter may be compromised by the ability of parasite substances to reach receptors on cells that they might ordinarily not be able to access due to permeability issues.

The review also makes extensive use of acronyms, some of which are defined on first use, others not.

Specific comments:

Line 19. Do the authors really mean to refer to the "immune system of the parasite"?

Line Line 42.  It is convention to refer to cells as T lymphocytes.

Line 66. What is meant by the term "executive" role?  A better term should be chosen. "Effector"?

Lines 101 to 109. The Figure legend describes numerous representatives with BH-3 only domain - but the list is not in agreement with the list in the figure itself.

Line 111. Is it the p53 domain that induces apoptosis or the protein?

In section 2.3, the authors describe the fate of keratinised cells. Are they really talking about cells of the gastrointestinal tract?

Line 229. Is the integrity of the mucous membrane really influenced by the differentiation of red blood cells (erythrocytes)?

Line 273-4.  The authors should find a better way of describing the anti-protozoal effects of metronidazole, than saying that it can "counter the pathogenicity".

Section 2.2. In talking about Giardia, the authors should avoid referring to the organism as both G. intestinalis and G. lamblia - one or the other and probably the former.

Line 304.  Excitation or excystation?

Line 306.  Invasiveness might be a term better reserved for describing the ability of organisms to penetrate into tissues rather than describing issues of infectivity.

Lines 334 to 336.  Cevallos et al, conducted a study, but what did they find?

Lines 346 to 347. The authors might like to comment on the biological significance of a shift in apoptotic coefficient of -0.1% to 0.2%.

Lines 379-80.  Surely it is Epidermal growth factor or EGF, not epithelial growth factor (EFG).

Line 414. The authors state that apoptosis is a crucial process for the survival of the parasite (Cryptosporidium spp.), but have failed to adequately describe why this is so.

Line 481 understating? or understanding?

Line 551 It should be villous atrophy or atrophy of villi.

Author Response

Reviewer 3

Whilst looking forward to reading this paper, I was ultimately somewhat disappointed. Partly this was down to a poor quality of english, but also since I felt an opportunity was missed. As a review article, there was the chance to provide an overview on why parasites might seek to promote, or to prevent apoptosis of epithelial cells, presumably since either might favour the biology and longevity of infections with the various organisms, or even to describe why control of apoptosis might be advantageous from the host's perspective, but such an overview was largely lacking. Instead the review rather reads as a list of studies and their findings. As an example, if villous atrophy can be mediated by substances released by N. brasiliensis that promote apoptosis, why would the parasite choose to do this when its presence in the gut is largely determined by its ability to coil its body around the villi, so that if they are lost, the parasite loses its ability to maintain its position in the gut. Such processes perhaps speak more of the host response to infection than to mechanisms the parasites use themselves to survive. It also illustrates the difficulty between determining the differences that occur between disease states associated with infection, and infections with the same parasite in the same host where disease does not arise. The latter situations arguably being more relevant to the natural host parasite relationship. The review would also have benefited from a clearer delineation of evidence from in vivo studies versus in vitro studies. Since the latter may be compromised by the ability of parasite substances to reach receptors on cells that they might ordinarily not be able to access due to permeability issues. The review also makes extensive use of acronyms, some of which are defined on first use, others not.

Authors: Thank you very much for the in-depth evaluation of the article, a positive opinion, as well as the critical remarks which provide important indications that helped us improve the quality of the article.

Reviewer 3: Line 19. Do the authors really mean to refer to the "immune system of the parasite"?

Authors: Thank you very much for the suggestions. This sentence was for the general conclusion. In the article, the authors mainly refer to changes in the host's body. Corrected according to the indications of Reviewer.

Reviewer 3: Line 42. It is convention to refer to cells as T lymphocytes.

Authors: Thank you for indicating the mistake. Corrected.

Reviewer 3: Line 66. What is meant by the term "executive" role? A better term should be chosen. "Effector"?

Authors: I agree with the Reviewer. The word "effector” is better than “executive”.

Reviewer 3: Lines 101 to 109. The Figure legend describes numerous representatives with BH-3 only domain - but the list is not in agreement with the list in the figure itself.

Authors: Thank you for indicating the wrong point. They are now included in Figure 1. This is our mistake.

Reviewer 3: Line 111. Is it the p53 domain that induces apoptosis or the protein?

Authors: Of course the protein. This is our mistake. Thank you for indicating the wrong point.

Reviewer 3: In section 2.3, the authors describe the fate of keratinised cells. Are they really talking about cells of the gastrointestinal tract?

Authors: We talking about human intestinal epithelial cells (IECs) in section 2.3. As other authors describe it: Induction of apoptosis before shedding of human intestinal epithelial cells. Grossmann J, Walther K, Artinger M, Rümmele P, Woenckhaus M, Schölmerich J. Am J Gastroenterol. 2002 Jun;97(6):1421-8. DOI: 10.1111/j.1572-0241.2002.05787.x.

We changed to the ,,human intestinal epithelial cells (IECs)" instead of ,, keratinised cells”.

Reviewer 3: Line 229. Is the integrity of the mucous membrane really influenced by the differentiation of red blood cells (erythrocytes)?

Authors: Thank you for indicating the additional points. This is our mistake. It was supposed to be ,,enterocytes”, not ,,erythrocytes”.

Reviewer 3: Line 273-4. The authors should find a better way of describing the anti-protozoal effects of metronidazole, than saying that it can "counter the pathogenicity".

Authors: We changed to ,, The study also revealed that the treatment with metronidazole can be an antiprotozoal drug of B. ratti. Consequently have a positive effect of the functioning of the epithelial membrane as the drug has been reported to induce Blastocystis cell death”.

Reviewer 3: Section 2.2. In talking about Giardia, the authors should avoid referring to the organism as both G. intestinalis and G. lamblia - one or the other and probably the former.

Authors: Corrected.

Reviewer 3: Line 304. Excitation or excystation?

Authors: Yes ,,excystation”.

Reviewer 3: Line 306. Invasiveness might be a term better reserved for describing the ability of organisms to penetrate into tissues rather than describing issues of infectivity.

Authors: Ok. Thank you for the suggestions.

Reviewer 3: Lines 334 to 336. Cevallos et al, conducted a study, but what did they find?

Authors: We've added the missing information.

Reviewer 3: Lines 346 to 347. The authors might like to comment on the biological significance of a shift in apoptotic coefficient of -0.1% to 0.2%.

Authors: We've added the missing information: ,,The experiment conducted by Troeger et al. (2007) demonstrated only a slight effect of G. intestinalis on apoptosis among patients infected with giardiasis. Presumably, apoptosis is triggered by direct contact of the protozoan or their products with the epithelium. Furthermore, the coefficient of apoptosis during intestinal giardiasis was found to have increased moderately to 0.2% as compared with the control – 0.1%. The results suggest that the even low-level release of proinflammatory cytokines such as TNF-α or IFN could lead to increased epithelial apoptosis inresponse to the chronic giardiasis. The apoptosis of intestinal epithelial cells can play an important role for epithelial barrier function, even if the increase is only moderately”.

Reviewer 3: Lines 379-80. Surely it is Epidermal growth factor or EGF, not epithelial growth factor (EFG).

Authors: Ok. Thank you for the suggestions. Corrected.

Reviewer 3: Line 414. The authors state that apoptosis is a crucial process for the survival of the parasite (Cryptosporidium spp.), but have failed to adequately describe why this is so.

Authors: We've added the missing information: ,,The programmed cell death is a crucial process both for the survival of the parasite, as well as for its pathogenicity. It results from the above that the apoptotic response of infected host cell is actively suppressed by parasite via upregulation of survivin, favoring by the parasite in order to complete its developmental stage. On the other hand Cryptosporidium may have evolved means to control host intestinal epithelial cells apoptosis to facilitate the establishment of infection in the newly exposed host. Furthermore, it can be concluded that infection with parasite is related to an immune response, and adaptation of the host organism to the antigens of this parasite. But the next observed changes in intestinal epithelial cells in rat indicate the defence against parasitic infestation”

Reviewer 3: Line 481 understating? or understanding? Line 551 It should be villous atrophy or atrophy of villi.

Authors: Corrected.

Round 2

Reviewer 1 Report

In the revised review manuscript by Kapczuk et al., the authors significantly improved the quality of the paper. Especially, the original photographs of various parasites are very impressive. In Figures 5, 7, and 8, 20x magnitude may not be necessary. The newly added Figure 2 is also easy to understand and helpful to readers. Therefore, I would like to recommend this review article for the publication in biomolecules. Congraturations!

Author Response

Thank you very much for the review, it helped us significantly improve the quality of our work, we appreciate your help.

Review

Thank you very much for the review, it helped us significantly improve the quality of our manuscript, we are very grateful for Reviewer’s help.

According to the reviewer's comment, we submitted the work to a native speaker. We send the certificate of our improved work.
